# Validation of an Inertial Sensor Algorithm to Quantify Head and Trunk Movement in Healthy Young Adults and Individuals with Mild Traumatic Brain Injury

**DOI:** 10.3390/s18124501

**Published:** 2018-12-19

**Authors:** Lucy Parrington, Deborah A. Jehu, Peter C. Fino, Sean Pearson, Mahmoud El-Gohary, Laurie A. King

**Affiliations:** 1Department of Neurology, Oregon Health & Science University, 3181 S.W. Sam Jackson Park Rd., Portland, OR 97239, USA; deb.jehu@ubc.ca (D.A.J.); peter.fino@utah.edu (P.C.F.); kingla@ohsu.edu (L.A.K.); 2VA Portland Health Care System, 3710 SW US Veterans Hospital Road, Portland, OR 97239, USA; 3Department of Health, Kinesiology and Recreation, University of Utah, 250 S 1850 E, Salt Lake City, UT 84112, USA; 4APDM Wearable Technologies, Portland, OR 97201, USA; seanp@apdm.com (S.P.); mahmoud@apdm.com (M.E.-G.); 5National Center for Rehabilitative Auditory Research (NCRAR), VA Portland Health Care System, 3710 SW US Veterans Hospital Road/P5, Portland, OR 97239, USA

**Keywords:** concussion, inertial motion units (IMUs), vestibular exercises, validation, motion capture

## Abstract

Wearable inertial measurement units (IMUs) may provide useful, objective information to clinicians interested in quantifying head movements as patients’ progress through vestibular rehabilitation. The purpose of this study was to validate an IMU-based algorithm against criterion data (motion capture) to estimate average head and trunk range of motion (ROM) and average peak velocity. Ten participants completed two trials of standing and walking tasks while moving the head with and without moving the trunk. Validity was assessed using a combination of Intra-class Correlation Coefficients (ICC), root mean square error (RMSE), and percent error. Bland-Altman plots were used to assess bias. Excellent agreement was found between the IMU and criterion data for head ROM and peak rotational velocity (average ICC > 0.9). The trunk showed good agreement for most conditions (average ICC > 0.8). Average RMSE for both ROM (head = 2.64°; trunk = 2.48°) and peak rotational velocity (head = 11.76 °/s; trunk = 7.37 °/s) was low. The average percent error was below 5% for head and trunk ROM and peak rotational velocity. No clear pattern of bias was found for any measure across conditions. Findings suggest IMUs may provide a promising solution for estimating head and trunk movement, and a practical solution for tracking progression throughout rehabilitation or home exercise monitoring.

## 1. Introduction

Individuals with mild traumatic brain injury (mTBI) suffering from impaired vestibular and ocular-motor impairments may be prescribed vestibular rehabilitation consisting of head and trunk movements during physical therapy. Vestibular rehabilitation typically includes gradual increases in the range of motion (ROM) and velocity of head movements and has shown promising improvements in the reduction of symptoms and greater overall function in mTBI patients [1,2,3]. While a physical therapist trains the patient in these exercises, a home exercise program is a vital part of vestibular rehabilitation therapy. Head and trunk movements can be compromised in this population, as the perceived head position relative to the trunk is generally impaired in individuals with vestibular pathologies [4]. Consequently, avoidance behavior and maladaptive strategies, such as limiting the head ROM and rotational velocity, may be used in an effort to minimize symptoms [4]. These subtle impaired movements, such as head and trunk velocity, are often not detected visually [5], and have the potential to interfere with successful rehabilitation. Thus, the ability to quantify and track these movements both within the clinic and during a home exercise program may be highly beneficial. 

Optical motion capture is commonly considered the gold-standard for tracking human movement. However, these motion capture systems are costly, can be limited by optical occlusion, and require a dedicated motion laboratory with specialists to collect, process and interpret the data. These factors make motion capture a suboptimal choice for clinical assessment in practice. Conversely, inertial measurement units (IMUs) provide a promising alternative due to the lower associated cost, simplified experimental set-up, and the ability to acquire data during everyday life, such as home monitoring [6].

IMUs have been shown to be a viable tool for measuring human motion across a wide range of activities and environments (for review across multiple domains see References [7,8,9,10,11]). While Duc and colleagues [12] have reported very good agreement with motion capture, and good retest reliability when measuring cervical ROM [12], it is unclear if IMUs can accurately estimate ROM and angular velocity during more complex movements. Additionally, the accuracy of IMU output can be subject to sensor placement [13], the developed algorithm, and how well it compensates for factors such as magnetic distortion and gyroscopic drift [14,15]. Further, it is important to validate an IMU algorithm across the range of measures of interest. 

Our lab is currently evaluating the use of IMUs within a standard vestibular focused home-exercise program involving head and trunk movements (ROM and peak rotational velocity) during standing and walking in individuals with mTBI. Prior to conducting the larger intervention, we wanted to determine the validity of the sensors that will be used. Therefore, the aim of this study was to determine the validity of IMUs to estimate head and trunk ROM and peak rotational velocity during head movements made when standing and walking, in comparison with three-dimensional motion capture data.

## 2. Methods

### 2.1. Participants

Five healthy young adults (3 females, mean (SD), age = 24.0 (2.3) years, height = 1.77 (0.12) m, mass = 77.4 (16.4) kg) and 5 individuals with mTBI (3 females, age = 36.4 (11.8) years, height = 1.74 (0.07) m, mass = 72.7 (11.2) kg, median (range), time since injury = 1.09 (0.43–4.38) years) participated in this study. Participants were included in the study if they were: (1) between 18 and 60 years old and (2) had zero to minimal cognitive impairment. In addition, the mTBI participants were included if they had a diagnosis of mTBI based upon Department of Defense criteria [16] and were still self-reporting symptoms related to their mTBI. Exclusion criteria included: (1) any musculoskeletal, neurological, or sensory problems that could explain balance deficits (not including the mTBI); (2) moderate to severe substance-use disorder within the past month (Diagnostic and Statistical Manual-5); (3) being in pain during the evaluation (≥7/10 by patient subjective report); (4) pregnancy; and (5) inability to abstain from medications that could impair balance for 24 h prior to testing. 

The study was conducted in accordance with the Declaration of Helsinki, and the protocol was approved by the Oregon Health & Science University Institutional Review Board (IRB #17206). All participants provided written informed consent prior to commencing testing.

### 2.2. Experimental Protocol

The testing protocol included four standing and four walking conditions (Table 1). Each of the conditions required either left and right (L/R) or up and down (U/D) continuous head movement. Two of the standing conditions were visual motion sensitivity (VMS) tasks and involved movement of the head and trunk en bloc. The walkway for the walking trials was 3.9 m long to stay within the optical field of the camera-based motion system. Participants completed two 30-second trials of each condition. All testing was completed at the Balance Disorders Laboratory at Oregon Health & Science University.

### 2.3. Equipment and Data Analysis

To collect inertial sensor data, two wearable IMUs; (APDM, Inc., Portland, OR, USA) were attached to the sternum and forehead of the participant. Each IMU included a tri-axial accelerometer (±6 g), gyroscope (±2000 °/s) and magnetometer (±6 gauss) that measured at a sampling frequency of 128 Hz. Moveo application (APDM, Inc.) was used to record the IMU data. The IMUs use wireless synchronization to ensure multiple units collect data with a precision of better than ±1 ms. Participants were also fitted with six reflective markers to collect simultaneous motion capture data. Markers were fixed to the forehead, the bilateral mandibular condyle of the head, the sternum, and the bilateral acromion process of the trunk. Motion capture data were collected using a 12-camera Motion Analysis system (120 Hz, Raptor-E, Motion Analysis Co., Santa Rosa, CA, USA) and processed using Cortex v6.2.3 (Motion Analysis, Co.). Motion capture was synchronized with the IMU recording using an APDM synchronization box. 

For each participant, a static trial was captured to define head and trunk segment position and orientation. This process allowed for each segment coordinate system to be rotated about the mediolateral axis, such that the anterior-posterior axis lies in the horizontal plane for the head and trunk in the static pose. A state space model and Kalman filter were used for sensor fusion between accelerometer, gyroscope, and magnetometer sensor data of the IMU [17]. Angular velocities were extracted from the head and trunk IMUs corresponding to rotations in the transverse plane (for L/R) and sagittal plane (for U/D). Angular displacement was calculated by integrating the angular velocity of the head in the intended direction (L/R or U/D). 

Optical data were filtered using a dual-pass second order Butterworth filter (6 Hz cut-off) and up-sampled to match the sampling rate of the IMU data. The head and trunk segments (defined in Table 2) used a right-hand coordinate system. Flexion, abduction, and axial rotation were decomposed using Euler angles. Segment angles for the rotations of interest were calculated and differentiated to estimate rotational velocities.

For both IMU and optical datasets, time series data were segmented into individual head turns, allowing the calculation of ROM and peak rotational velocity. For the walking and tandem walking trials, only the straight walking segments were included. Portions of the trial when participants were turning at the ends of the walking path were removed. Turns were detected using a threshold turn angle greater than 45° and a peak turn velocity greater than 15 degrees per second [18].

### 2.4. Statistical Analysis

Validity of the lMU-based algorithm was assessed using a combination of Intra-class Correlation Coefficients (ICC(A,1)) for assessing absolute agreement, root mean square error (RMSE), as well as percent error between the IMU and criterion value of the motion capture data. Interpretation of ICC values were based on Koo and Li [19], with the following cut-offs: poor <0.5; moderate = 0.5–0.75; good = 0.75–0.9; and excellent >0.9. Bland-Altman plots [20] were also examined to gain an understanding of any patterns relating to bias.

## 3. Results

Representative time series data for walking with head turns L/R and walking with head turns U/D are displayed for one healthy participant (Figure 1) and one participant with mTBI (Figure 2).

Correlation, RMSE, and percent error results for the head and trunk are presented in Table 3 and Table 4, respectively. The IMU data strongly represented the criterion motion capture data for head ROM and peak rotational velocity across all conditions (ICC(A,1) > 0.9). RMSE across conditions remained low for head ROM but increased in the walking L/R and tandem walking L/R conditions for peak rotational head velocity. Despite the higher RMSE, the percent error for the head remained low across all conditions (<5%).

Intra-class Correlations Coefficients between the IMU and motion capture data for the trunk were stronger in the L/R direction (ROM ICC(A,1) > 0.9; peak rotational velocity ICC(A,1) > 0.9), than in the U/D direction (ROM ICC(A,1) = 0.580 to 0.907; peak rotational velocity ICC(A,1) = 0.436 to 0.787) across conditions. The reduced strength of the relationship is also mirrored in the RMSE and percent error scores for the U/D motions.

Bland-Altman plots did not indicate any clear patterns of bias. Examples are provided below for the head ROM and peak rotational velocity during walking with head turns (Figure 3).

## 4. Discussion

In this study we investigated the validity of IMUs to detect and measure head and trunk ROM and peak rotational velocity during a set of commonly prescribed vestibular rehabilitation tasks. Our findings suggest excellent validity for the IMU system when capturing head movements in both the L/R and U/D conditions and excellent validity capturing trunk movements in the L/R conditions. Inertial sensors showed moderate to excellent ability to estimate trunk ROM in the U/D conditions and was moderate to good at capturing peak rotational velocity in U/D conditions; except during tandem walking which showed poor agreement. The excellent agreement found here for head motion is consistent with previous findings [12], who showed excellent agreement for cervical angles collected using inertial sensors on the head and neck. This work also extends prior research, by identifying that IMUs can accurately estimate the ROM and peak turning velocity during both standing and locomotor tasks.

Automatically characterizing head and trunk movements during routinely prescribed vestibular exercises using IMUs is an innovative approach that will allow a more sensitive and objective analysis of progression during vestibular rehabilitation. In people with mTBI, smaller and slower head movements during performance tasks have been reported [5] but such movements are not easily quantified with the naked eye and may not be perceived by the patient performing the exercise. Quantifying such information with IMUs could inform both the treating physical therapist and, with time, the patient themselves by providing immediate feedback on velocity and quality of performance. 

Despite the good agreement between IMU and motion capture systems, we believe some of the estimation errors might be attributed to a misalignment of the IMU frame relative to the anatomical axes of rotation. When the IMUs are attached to different body segments, they are not perfectly aligned with the segments’ main axes of rotation. To estimate this misalignment, we asked the study participants to remain stationary in a neutral pose for about three seconds at the beginning of the recording. This information was then used to realign the sensors’ data for analysis, using matrix rotation, before calculating the joint metrics. While this addressed the misalignment of the sensors relative to the anatomical axes, it assumed the participant could both remain stationary and adopt a truly neutral initial pose. This was not always true for every participant, and we hypothesize that this contributed to the larger errors observed in a few of the subjects.

Although 3D motion capture is commonly classified as a gold-standard measurement, it is possible that reduced agreement in some cases could partially be a function of the motion capture methods implemented in this study. Firstly, the accuracy of motion capture systems can decrease as the capture volume increases [21]. Despite using a 12-camera system it is possible that the size of our capture area, elongated to collect the tandem and walking trials, played a role in the reduced agreement between the two systems. Another source of disagreement between the inertial and optical systems could be attributed to skin motion artifact and muscle movements—a known issue [22] with systems that use markers attached to the body. Similarly, skin artifacts can also influence the inertial sensor measurements resulting in potential orientation changes. These orientation changes may produce joint metric estimates that are biomechanically unlikely and lead to a disagreement between the systems.

### 4.1. Clinical Implications

Inertial measurement technologies enable clinicians to capture patients’ movement during unconstrained activities—whether in the clinic or during daily living within their home environment. As these tools become more accessible to therapists and patients, the information they provide has the potential to help improve diagnosis and recommendations for therapeutic interventions or rehabilitation strategies. Additionally, they can facilitate the collection of clinical trial outcome measures in most outpatient clinics, rather than specialized laboratory settings. Nonetheless, we note that IMU-based systems are not a replacement for clinical experience, but rather a tool that can complement clinical judgment. Furthermore, future studies are needed to determine the feasibly of using these systems during rehabilitation programs to monitor compliance and the progression of exercise intensity.

### 4.2. Limitations

There are some limitations to our study that should be noted. First, validation data are limited to the exercises of interest to our group. These vestibular exercises were chosen based on their common use and prescription to persons with ongoing balance problems after mTBI. Nonetheless, validation data in this study may not be generalizable to more dynamic tasks involving multiple planes of motion. Any estimation of trunk and head ROM and peak rotational velocities in more complex movements may, therefore, require further validation. Second, as accuracy of IMU output can be subject to sensor placement [13], it is possible that a different placement or a combined placement of sensors (i.e., an IMU on both the sternum and the lumbar spine), could provide a more accurate estimation of trunk motion. Third, our capture volume for optical motion data was limited, resulting in occasional marker occlusions when participants turned around the ends of our capture volume. Gap filling of missing optical markers and the resulting underestimation of head angles may be responsible for the fluctuations in error coincident with the turns. Fourth, errors in the trunk ROM and peak velocity may have been minimized had the trunk been defined differently. For U/D head motions, in particular, it is plausible that movement artifact occurred with neck movement pulling on the skin at the top of the sternum where one marker was placed. Finally, the IMU algorithm required the participants to stand in a static position so that the orientation of the sensors could be defined. This assumes that subjects can maintain a neutral stationary position, and further investigation into the effect of this potential issue is warranted.

## 5. Conclusions

Our findings suggest that it is feasible to use IMUs to measure head and trunk movement and provide metrics that are clinically important. The system is portable, unobtrusive, and easy to use. These features make such systems well suited for use in the clinic to detect and characterize head and trunk movement during routine and standard vestibular rehabilitation. This study provides an initial step towards the implementation of IMUs to provide clinically meaningful information to physical therapists treating patients with imbalance after mTBI.

## Figures and Tables

**Figure 1 sensors-18-04501-f001:**
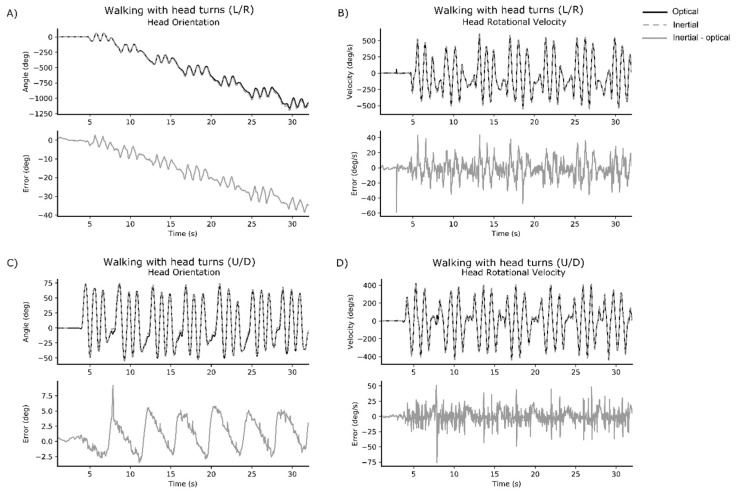
Representative time series data from one healthy participant. Each subplot provides the optical signal (black solid line) and IMU signal (grey dashed line) overlaid in the upper figure, and the error between optical and IMU signal below (grey solid line). Subplots represent: (**A**) head orientation for walking with head turns (L/R); (**B**) peak rotational velocity of the head for walking with head turns (L/R); (**C**) head orientation for walking with head turns (U/D); and (**D**) peak rotational velocity of the head for walking with head turns (U/D).

**Figure 2 sensors-18-04501-f002:**
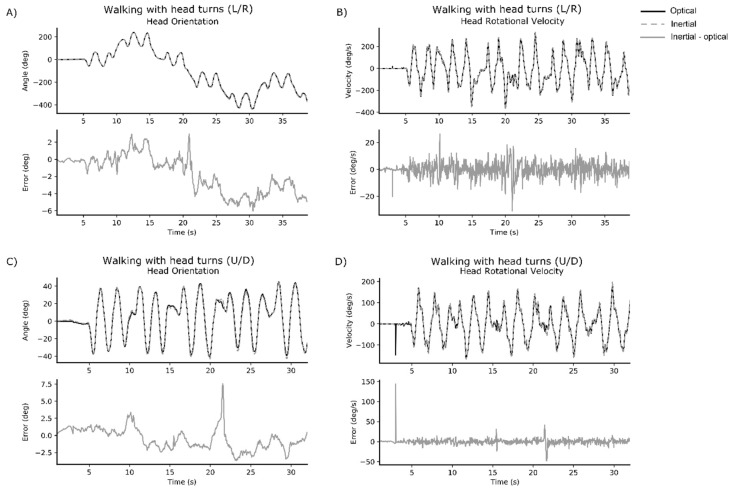
Representative time series data from one participant with mTBI. Each subplot provides the optical signal (black solid line) and IMU signal (grey dashed line) overlaid in the upper figure, and the error between optical and IMU signal below (grey solid line). Subplots represent: (**A**) head orientation for walking with head turns (L/R); (**B**) peak rotational velocity of the head for walking with head turns (L/R); (**C**) head orientation for walking with head turns (U/D); and (**D**) peak rotational velocity of the head for walking with head turns (U/D).

**Figure 3 sensors-18-04501-f003:**
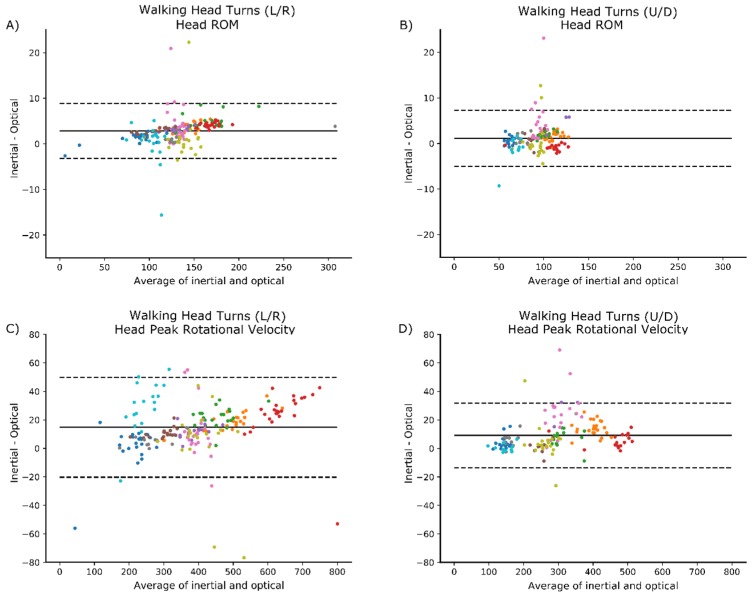
Bland-Altman plots for walking conditions. Each participant is represented by a different color within the plot. (**A**) ROM, walking with head turns (L/R); (**B**) ROM, walking with head turns (U/D); (**C**) peak rotational velocity, walking with head turns (L/R); and (**D**) peak rotational velocity, walking with head turns (U/D). Solid lines represent the mean difference, and dashed lines represent ±1.96 × SD.

**Table 1 sensors-18-04501-t001:** Description of standing and walking conditions.

Standing	Standing VMS	Walk	Tandem Walk
Standing with feet together while turning the head L/R. 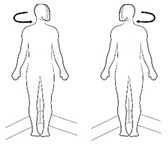	Standing with feet together while moving head and trunk together. Participants held their dominant arm straight out in front of the body, fixed their eyes on their thumb, and concurrently moved the head, trunk, and arms en bloc L/R. 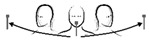	Participants walked continuously at a comfortable pace while turning the head L/R. 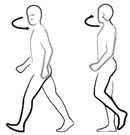	Participants walked in tandem at a comfortable pace while turning the head L/R. 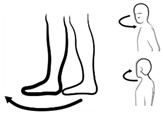
Standing with feet together while moving the head U/D. 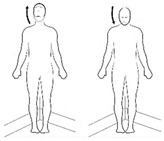	Standing with feet together while moving head and trunk together. Participants held their dominant arm straight out in front of the body, fixed their eyes on their thumb, and concurrently moved the head, trunk, and arms en bloc U/D. 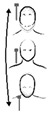	Participants walked continuously at a comfortable pace while moving the head U/D. 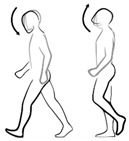	Participants walked in tandem at a comfortable pace while moving the head U/D. 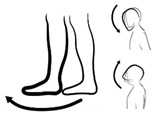

Note: VMS = visual motion sensitivity; L/R = Left and right; U/D = Up and down.

**Table 2 sensors-18-04501-t002:** Head and trunk segment definitions.

Segment	Origin/Axes	Definition
Head	Origin	Midpoint between right and left mandibular condyle markers
	Mediolateral axis	Projected from the origin to the right mandibular condyle
	Anterior posterior axis	Projected from the origin to the forehead marker
	Vertical axis	Projected from the origin, orthogonal to the AP and ML axes
Trunk	Origin	Midpoint between right and left acromion process
	Mediolateral axis	Projected from the origin to the right acromion process
	Anterior posterior axis	Projected from the origin to the sternum marker
	Vertical axis	Projected from the origin, orthogonal to the AP and ML axes

**Table 3 sensors-18-04501-t003:** Validity results comparing IMU to motion capture for head range of motion (ROM) and peak rotational velocity (ω_p_).

	ICC(A,1)		RMSE		% Error	
Condition	ROM	ω_p_	ROM	ω_p_	ROM	ω_p_
Standing L/R	0.993	0.994	2.39	11.58	−1.5	−1.6
Standing U/D	0.997	0.991	1.36	9.59	−0.4	−2.3
Standing VOR L/R	0.992	0.994	3.78	6.73	−1.8	−0.3
Standing VOR U/D	0.998	0.992	1.29	5.55	0.02	−2.0
Walking L/R	0.991	0.986	3.55	20.26	−2.0	−4.0
Walking U/D	0.985	0.991	2.71	12.52	−1.2	−3.2
Tandem walking L/R	0.994	0.987	2.93	17.04	−1.9	−4.4
Tandem walking U/D	0.985	0.988	3.11	10.84	−0.4	−2.3
Mean across conditions	0.992	0.990	2.64	11.76	−1.1	−2.5
SD across conditions	0.005	0.003	0.86	4.61	0.8	1.3

**Table 4 sensors-18-04501-t004:** Validity results comparing IMU to motion capture for trunk ROM and peak rotational velocity.

	ICC(A,1)		RMSE		% Error	
Condition	ROM	ω_p_	ROM	ω_p_	ROM	ω_p_
Standing L/R	0.986	0.960	0.59	2.31	−6.0	−6.9
Standing U/D	0.580	0.815	1.82	5.41	−29.3	−11.7
Standing VOR L/R	0.985	0.997	3.55	4.16	1.9	0.6
Standing VOR U/D	0.907	0.787	6.49	12.71	19.2	5.3
Walking L/R	0.997	0.976	1.17	5.47	0.8	−6.3
Walking U/D	0.843	0.746	2.60	12.08	−13.2	−17.1
Tandem walking L/R	0.998	0.976	0.96	3.69	−1.5	−3.5
Tandem walking U/D	0.639	0.436	2.66	13.14	−2.9	0.2
Mean across conditions	0.867	0.837	2.48	7.37	−3.9	−4.9
SD across conditions	0.169	0.189	1.78	4.20	13.8	7.2

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
