# Peer review of "Validation of an Inertial Sensor Algorithm to Quantify Head and Trunk Movement in Healthy Young Adults and Individuals with Mild Traumatic Brain Injury"

_sensors, 2018, doi:10.3390/s18124501_

Round 1

Reviewer 1 Report

The main idea:

An IMU motion capture system is available to monitor young adults and patients with mTBI. The validity is shown by comparing the processed sensor data with data of an optical motion capture system. It is relatively easy to use in rehabilitation settings because it is portable and no there is no need to support from a motion capture expert.

Comments:

Concerning the language, readability is good and well understandable. The structure is clear.

Page 2, Line 60:

Some important examples/citations should be introduced additionally:

In sports:

M. Brodie, A. Walmsley, and W. Page, “Fusion motion capture: A prototype system using inertial measurement units and GPS for the biomechanical analysis of ski racing,” Sports Technol., vol. 1, no. 1, pp. 17–28, Jun. 2008.

Concerning Motor Learning:

A. O. Effenberg, U. Fehse, G. Schmitz, B. Krueger, and H. Mechling, “Movement sonification: Effects on motor learning beyond rhythmic adjustments,” Front. Neurosci., vol. 10, p. 219, May 2016.

Gait Analysis with head movement:

T.-H. Hwang, J. Reh, A. O. Effenberg, and H. Blume, “Real-time gait analysis using a single head-worn inertial measurement unit,” IEEE Trans. Consumer Electronics, vol. 64, no. 2, pp. 240–248, May 2018.

In Operation:

C. He, P. Kazanzides, H. T. Sen, S. Kim, and Y. Liu, “An inertial and optical sensor fusion approach for six degree-of-freedom pose estimation,” Sensors, vol. 15, no. 7, pp. 16448–16465, 2015.

Fall protection with IMU-based smartphone:

A. J. A. Majumder, P. Saxena, and S. I. Ahamed, “Your walk is my command: Gait detection on unconstrained smartphone using IoT system,” in Proc. IEEE 40th Annu. COMPSAC, Atlanta, GA, USA, 2016,

pp. 798–806.

Page 4,

Table 2 : The second segment should be “Trunk.”

Line 121: … velocity of greater … -> … velocity greater …..

Line 131: … from one participant are …..

Is the participant healthy or with mTBI? From my perspective, it seems that the representative data are from a healthy participant. How about showing data from participants with mTBI and comparing to healthy participants. Then you can show the system can suitable for participants with mTBI.

Page 5,

Figure 1.

This figure is related to two conditions out of four. Please mention the conditions.

Line. 133: Appropriate legend of lines is needed.

Please mention which is the solid line and the dashed line. Black and signal is not appropriate information. 

(A) From other plots, participants seem to turn every 5 seconds when they reach the end of the 3.9 m walking lane. Please explain why the plot (A) is free from turning of the body. 

For Figure 1 (A) and (C), the plot shows an angle curve, but not the ROM. Range of movement can be measured with a certain temporal window.

(C) Please explain why error (deg) is higher and abruptly changing when the participant (might) turn. Is there any effect when participants reach the end of the stage of the optical system?

Page 6,

Figure 2.

For the intuitive comparison, you would better set the same sales of horizontal and vertical axes of subplots. And, how about changing the position subplot B and C? You can easily compare Head ROMs between L/R and U/D condition. You can also compare Head peak rotational velocities between L/R and U/D conditions.

Author Response

We thank each of the reviewers for their thoroughness and constructive feedback. Response to reviewer comments and all changes that have been made are documented below, and are highlighted in the revised manuscript in yellow.

Page 2, Line 60:

Some important examples/citations should be introduced additionally:

We appreciate the suggestion of the reviewer to include additional examples. Given the use of inertial sensors across multiple domains, we believe that referring to reviews of IMU use in multiple different areas (e.g. clinical and sports) would provide readers with a broader scope. Referral is now made within the introduction.

Page 4,

Table 2: The second segment should be “Trunk.”

Amended

Line 121: … velocity of greater … -> … velocity greater …..

Amended

Line 131: … from one participant are …..

Is the participant healthy or with mTBI? From my perspective, it seems that the representative data are from a healthy participant. How about showing data from participants with mTBI and comparing to healthy participants. Then you can show the system can suitable for participants with mTBI.

The participant shown in Figure 1 was a healthy control. We agree that showing data from an mTBI would also be useful in order to show suitability for using in patients with mTBI and have generated a second sub-plotted figure, now Figure 2.

Page 5,

Figure 1.

This figure is related to two conditions out of four. Please mention the conditions.

We have now indicated that the figure represents data from walking with head turns L/R and walking with head turns U/D.

Line. 133: Appropriate legend of lines is needed.

A legend has been added to the figure.

Please mention which is the solid line and the dashed line. Black and signal is not appropriate information. 

This has been amended to say “Each subplot provides the optical signal (black solid line) and IMU signal (grey dashed line) overlaid in the upper figure”.

(A) From other plots, participants seem to turn every 5 seconds when they reach the end of the 3.9 m walking lane. Please explain why the plot (A) is free from turning of the body. 

The reviewer is correct in that the Figure 1.A was a standing trial rather than a walking one. Thank you for highlighting this. We have amended this within the figure. We note that the walking L/R head orientation time series is slightly less intuitive than the U/D condition or the standing trials. This is because with every turn there is effectively 180 degrees added or subtracted from the signal and the signal is unraveled. We note that in Figure 2 (now added to revised manuscript) that this participant turns different directions, while the participant in Figure 1 turns only one direction throughout the trial. This results in a different pattern over the course of the time series. Nonetheless the maxima and minima per head turn can be identified throughout.

For Figure 1 (A) and (C), the plot shows an angle curve, but not the ROM. Range of movement can be measured with a certain temporal window.

The figure caption has been reworded from ROM to ‘orientation’.

(C) Please explain why error (deg) is higher and abruptly changing when the participant (might) turn. Is there any effect when participants reach the end of the stage of the optical system?

Increased error in head orientation in this condition (walking with head turns U/D) for this participant may be related to the missing optical markers at each turn. Our limited capture volume required turns at the end of the walkway. In some instances, the optical markers were occluded or traveled outside our capture volume. In these cases, the optical marker data was gap filled using splines, and these spline fills likely underestimated the rotation of the head. This would explain the relatively large shifts in error coincident with the turns. We have added a statement addressing this limitation to the manuscript.

Importantly, while we provide time series data as an example, these turning segments are not included within the statistical analysis of ICC, RMSE and %error. As identified within our methods, only the straight walking segments are included in the analysis (page 8).

Page 6,

Figure 2. For the intuitive comparison, you would better set the same sales of horizontal and vertical axes of subplots. And, how about changing the position subplot B and C? You can easily compare Head ROMs between L/R and U/D condition. You can also compare Head peak rotational velocities between L/R and U/D conditions.

We have adjusted the figure axes and the position of the subplots as per the reviewer’s suggestion.

Reviewer 2 Report

The authors present a validation of IMUs for rehabilitation of mild TBI patients. A study with 10 participants was performed and IMU output was compared against a motion capturing system as gold standards.

While the article was well written and in general a good and refreshingly concise read, I have two major points of criticism:

1. No ethical approval or study registration was reported while part of the participants in the trial are patients and the protocol reads much like the description of a clinical trial. In this regard I direct the authors to the author instructions, section research and publication ethics: https://www.mdpi.com/journal/sensors/instructions#ethics

Also, why does everybody exclude pregnant women? It seems unnecessary in this context.

2. What is the novelty of the approach over Duc et al. 2014 (A wearable inertial system to assess the cervical spine mobility: Comparison with an optoelectronic-based motion capture evaluation)? The articles seem to have similar results as well as similar outcome.

Minor remarks:

* It would be nice to provide the manuscript for citation 5 alongside for the review as this paper was not accessible or findable.

* Why was the sternum chosen as fixed-point for the lower IMU instead of a position on the back? Shouldn't breathing effect the measures at least in the sagittal plane? This could explain the lower ICC in the U/D condition. (Figure 1 C error looks a bit like a breathing pattern with 12 breaths per minute although head orientation should not be influenced by breathing according to Table 2)

* The second head in the segment column of Table 2 should probably be trunk

Author Response

We thank each of the reviewers for their thoroughness and constructive feedback. Response to reviewer comments and all changes that have been made are documented below, and are highlighted in the revised manuscript in yellow.

While the article was well written and in general a good and refreshingly concise read, I have two major points of criticism:

1. No ethical approval or study registration was reported while part of the participants in the trial are patients and the protocol reads much like the description of a clinical trial. In this regard I direct the authors to the author instructions, section research and publication ethics: https://www.mdpi.com/journal/sensors/instructions#ethics

We apologize for this oversight. The following statement and Institutional Review Board identification number have been inserted into the manuscript (page 5):

The study was conducted in accordance with the Declaration of Helsinki, and the protocol was approved by the Oregon Health & Science University Institutional Review Board (IRB #17206). All participants provided written informed consent prior to commencing testing.

Also, why does everybody exclude pregnant women? It seems unnecessary in this context.

Postural control and dynamic balance can be effected in pregnant women. While we were not specifically measuring balance in the context of this validation study, the exercises performed in each condition tested are specific to vestibular rehabilitation and therefore known to challenge the vestibular and oculomotor system. We did not want potential confounding from this population.

Butler, E. E., Colón, I., Druzin, M. L., & Rose, J. (2006). Postural equilibrium during pregnancy: decreased stability with an increased reliance on visual cues. American journal of obstetrics and gynecology, 195(4), 1104-1108.

Inanir, A., Cakmak, B., Hisim, Y., & Demirturk, F. (2014). Evaluation of postural equilibrium and fall risk during pregnancy. Gait & posture, 39(4), 1122-1125.

McCrory, J. L., Chambers, A. J., Daftary, A., & Redfern, M. S. (2010). Dynamic postural stability during advancing pregnancy. Journal of biomechanics, 43(12), 2434-2439.

2. What is the novelty of the approach over Duc et al. 2014 (A wearable inertial system to assess the cervical spine mobility: Comparison with an optoelectronic-based motion capture evaluation)? The articles seem to have similar results as well as similar outcome.

In comparison to Duc et al. our investigation observed head and trunk motion while completing both standing and locomotor (walking and tandem walking) tasks. In addition, we present both range of motion and peak turn velocity. We believe that this study provides support for previous work, as well as new information, and have now made mention of this within the introduction and discussion.

Minor remarks:

* It would be nice to provide the manuscript for citation 5 alongside for the review as this paper was not accessible or findable.

I apologize for the inability to find this article. I had incorrectly entered this into EndNote, which is likely why it could not be found. This has been corrected in the manuscript, and I have also attached the article with this revision.

* Why was the sternum chosen as fixed-point for the lower IMU instead of a position on the back? Shouldn't breathing effect the measures at least in the sagittal plane? This could explain the lower ICC in the U/D condition. (Figure 1 C error looks a bit like a breathing pattern with 12 breaths per minute although head orientation should not be influenced by breathing according to Table 2)

Placement of the sensor on the sternum/ chest area was based off initial design and commercialization of the APDM sensors with Mobility Lab, where placement of the IMU on the sternum was affective and accurate for the original applications. APDM is currently developing new straps to better work for joint angle estimation including new design for the lumbar and sternum sensor straps. However, as we are running a larger rehabilitation trial, it was necessary to assess the validity of sensors using the same attachment straps and placement that is being used within the larger study.

Re: lower ICC in the U/D condition:

Although there is a possibility that the placement of the sensor on the sternum is responsible for reduced ICC etc., if breathing was the source of reduced accuracy, we would expect to see a lower accuracy in both the L/R and U/D conditions. Comparatively, ICC, RMSE and %error were much better in the L/R than U/D conditions.

Rather, we speculate that the source of error in the trunk data of the U/D conditions may be related to surface movement artifact. When the head nods up and back it pulls on skin covering the jugular notch and has the potential to move the sternum marker up and down with each up and backward nod of the head. We have now made reference to this in the limitations section (page 16).

* The second head in the segment column of Table 2 should probably be trunk

This has been corrected.

Reviewer 3 Report

The paper describes validation of use of IMU system for quantifying head movements for vestibular rehabilitation. IMU system is validated using optical system. Signals acquired with IMUs and reference optical system are compered.

Paper is in general well written, methodology is well explained and executed, results of the measurements are adequately and clearly presented.

I had some questions that were actually clarified in the section Clinical implications and section Limitations.

My biggest reservation is that the paper and study presented in the paper does not bring enough novelty. Literature presenting various uses of IMUs is quite was and I do not find the finding that: IMUs may provide a promising solution for estimating head and trunk movement, and a practical solution for tracking progression throughout rehabilitation or home-exercise monitoring is not particularly novel. This is what should actually be tested. Presented study is well executed; however, it is also very basic. Study is a good basis for more thorough study of use of the IMUs system in vestibular rehabilitation.

I would suggest that for such study system is validated with methodology that is actually used for training and assessment used by professional therapist. This study should also be conducted over several sessions so that also assessment using IMUs and traditional methods used in assessment by professional therapist can be compared.

This study essentially shows that IMUs are accurate system compared to optical system, which indeed is de-facto standard for validation of IMUs system, however the study does not show actual usefulness of the IMU system in therapy for training or assessment.

Author Response

We thank each of the reviewers for their thoroughness and constructive feedback. Response to reviewer comments and all changes that have been made are documented below, and are highlighted in the revised manuscript in yellow.

My biggest reservation is that the paper and study presented in the paper does not bring enough novelty. Literature presenting various uses of IMUs is quite was and I do not find the finding that: IMUs may provide a promising solution for estimating head and trunk movement, and a practical solution for tracking progression throughout rehabilitation or home-exercise monitoring is not particularly novel. This is what should actually be tested. Presented study is well executed; however, it is also very basic. Study is a good basis for more thorough study of use of the IMUs system in vestibular rehabilitation.

I would suggest that for such study system is validated with methodology that is actually used for training and assessment used by professional therapist. This study should also be conducted over several sessions so that also assessment using IMUs and traditional methods used in assessment by professional therapist can be compared.

We agree with the reviewer that this study provides a basis for a more thorough study of the use of the IMU system in vestibular rehabilitation, and we appreciate that the novelty of this article may seem limited in comparison with a more substantial investigation in that area.

The current study is the first progression within a larger investigation that evaluates the use of wearable sensors during a home-based exercise program. The full study has two primary aims – one assessing the timing of rehabilitation, and the second assessing whether wearing sensors while completing home-exercises can improve rehabilitation outcomes following mTBI.

We have now made it more explicit within the introduction the gap that is filled, and that this study is an initial step as part of a broader study investigating the use of wearable sensors for home-exercises (page 4).

This study essentially shows that IMUs are accurate system compared to optical system, which indeed is de-facto standard for validation of IMUs system, however the study does not show actual usefulness of the IMU system in therapy for training or assessment.

Importantly, our team wished to know that the sensors were tracking the desired movements correctly. Specifically, the larger study is a randomized control trial that aims to collect data (ROM and peak velocity) from these sensors during the home-exercise rehabilitation program. Throughout the larger study, Physical Therapists will be administering the exercise program, and measuring motion with the sensors as well measuring rotation (with a goniometer) and flexion-extension (with an inclinometer) for comparison. Physical Therapists will also report on their experiences using the system.

We believe validating the sensors for the movements we aim to analyze throughout the larger study is a necessary foundation step that allows our team to have confidence in the data we are hoping to track. Further, we feel that publishing this data is appropriate to provide information for persons wishing to know more about the tools used in the upcoming investigations.

Round 2

Reviewer 3 Report

I have read the response from the authors and I still believe that the verification method is very basic. Only study that compares the estimations of the angles was conducted in conditions that might not be identical to those of the more thorough study that will be conducted with training scenario. Might comments were addressed only in form of additional explanation, no things with methodology or additional experiments were made. Therefore, I cannot comment it.

In short, I can only repeat my previous reservation: although the presented study is properly executed, it is very basic and does not really bring any novel insights. Main conclusion is that there is agreement in IMU results and results from reference system. But this has already been shown multiple times in various experiments. A proper validation would be to show that IMUs are actually useful for real application, that is for real training scenario. My opinion is that this results do not yet have enough significance.

 I leave decision to editors, which need to determine if the current results are significant enough for publication in the journal.

Author Response

Validation of an inertial sensor algorithm to quantify head and trunk movement in healthy young adults and individuals with mild traumatic brain injury (sensors-383537)

Response to Reviewer 3 (Round 2)

Comments and Suggestions for Authors

I have read the response from the authors and I still believe that the verification method is very basic. Only study that compares the estimations of the angles was conducted in conditions that might not be identical to those of the more thorough study that will be conducted with training scenario. Might comments were addressed only in form of additional explanation, no things with methodology or additional experiments were made. Therefore, I cannot comment it.

In short, I can only repeat my previous reservation: although the presented study is properly executed, it is very basic and does not really bring any novel insights. Main conclusion is that there is agreement in IMU results and results from reference system. But this has already been shown multiple times in various experiments. A proper validation would be to show that IMUs are actually useful for real application that is for real training scenario. My opinion is that this results do not yet have enough significance.

I leave decision to editors, which need to determine if the current results are significant enough for publication in the journal.

The reviewer’s concerns in both the first and second round of reviews relate to the novelty of the study. It is true that work has previously been conducted to evaluate the use of IMUs for estimating neck motion (e.g. Theobald et al., 2012; Duc et al., 2014). Nonetheless, our study differs from this work in multiple ways, including 1) the sensors; 2) the algorithms; 3) the more complex movements that are conducted (dynamic, complex vestibular rehabilitation based head and trunk movements); 4) the population; and 5) the measures that are collected. We also highlighted within our response that this validation was a foundation step within a multistage study that is being conducted.

The need for further validation of inertial sensors has been previously documented. This is especially the case when sensors will be used outside of the laboratory for assessment, and where more complex human movements are being measured. Specifically, one systematic review concluded that while IMUs can offer an accurate and reliable method to study human motion, the degree of accuracy and reliability is site and task specific (Cuesta-Vargas et al, 2013). Furthermore, another systematic review assessing more complex movements (e.g. sports related) concluded that given contrasting results across multiple validation studies, further research is required to validate the ability of IMUs to be used in more complex activities (e.g. Chambers et al, 2015).

As we address in our initial response to reviewer 3, this validation forms the foundation of work within a 4-year randomized control trial (Sensory integration balance deficits in complex mTBI: Can early initiation of rehabilitation with wearable sensor technology improve outcomes? DOD award number W81XWH-17-1-0420). This validation study was conducted with the purpose of validating the specific tasks that will be used throughout the larger study to track movements during home exercises and physical therapy sessions.

Inertial measurement units are still far from being used in the clinic.  We believe our larger study, which is using IMUs for vestibular rehabilitation after mTBI will be an important step to translating this technology for clinicians. For this reason we believe the validation of our algorithm and specific tasks for vestibular rehabilitation are important to publish.

Original comments and response to Reviewer 3

Reviewer 3: Comments and Suggestions for Authors

My biggest reservation is that the paper and study presented in the paper does not bring enough novelty. Literature presenting various uses of IMUs is quite was and I do not find the finding that: IMUs may provide a promising solution for estimating head and trunk movement, and a practical solution for tracking progression throughout rehabilitation or home-exercise monitoring is not particularly novel. This is what should actually be tested. Presented study is well executed; however, it is also very basic. Study is a good basis for more thorough study of use of the IMUs system in vestibular rehabilitation.

I would suggest that for such study system is validated with methodology that is actually used for training and assessment used by professional therapist. This study should also be conducted over several sessions so that also assessment using IMUs and traditional methods used in assessment by professional therapist can be compared.

We agree with the reviewer that this study provides a basis for a more thorough study of the use of the IMU system in vestibular rehabilitation, and we appreciate that the novelty of this article may seem limited in comparison with a more substantial investigation in that area.

The current study is the first progression within a larger investigation that evaluates the use of wearable sensors during a home-based exercise program. The full study has two primary aims – one assessing the timing of rehabilitation, and the second assessing whether wearing sensors while completing home-exercises can improve rehabilitation outcomes following mTBI.

We have now made it more explicit within the introduction the gap that is filled, and that this study is an initial step as part of a broader study investigating the use of wearable sensors for home-exercises (page 4).

This study essentially shows that IMUs are accurate system compared to optical system, which indeed is de-facto standard for validation of IMUs system, however the study does not show actual usefulness of the IMU system in therapy for training or assessment.

Importantly, our team wished to know that the sensors were tracking the desired movements correctly. Specifically, the larger study is a randomized control trial that aims to collect data (ROM and peak velocity) from these sensors during the home-exercise rehabilitation program. Throughout the larger study, Physical Therapists will be administering the exercise program, and measuring motion with the sensors as well measuring rotation (with a goniometer) and flexion-extension (with an inclinometer) for comparison. Physical Therapists will also report on their experiences using the system.

We believe validating the sensors for the movements we aim to analyze throughout the larger study is a necessary foundation step that allows our team to have confidence in the data we are hoping to track. Further, we feel that publishing this data is appropriate to provide information for persons wishing to know more about the tools used in the upcoming investigations.